# Pitfalls and Tips in the Assessment of Aortic Stenosis by Transthoracic Echocardiography

**DOI:** 10.3390/diagnostics13142414

**Published:** 2023-07-19

**Authors:** Grazia Canciello, Shabnam Pate, Anna Sannino, Felice Borrelli, Gaetano Todde, Paul Grayburn, Maria-Angela Losi, Giovanni Esposito

**Affiliations:** 1Department of Advanced Biomedical Sciences, Federico II University, 80131 Naples, Italy; grazia.canciello@hotmail.it (G.C.); anna.sannino@unina.it (A.S.); felice.borrelli@yahoo.it (F.B.); gaetano.todde@virgilio.it (G.T.); espogiov@unina.it (G.E.); 2Division of Cardiology, Baylor Scott & White Research Institute, Plano, TX 75204, USA; shabnam.pate@bswhealth.org (S.P.); paul.grayburn@bswhealth.org (P.G.)

**Keywords:** aortic stenosis, echocardiography

## Abstract

Aortic stenosis (AS) is a valvular heart disease that significantly contributes to cardiovascular morbidity and mortality worldwide. The condition is characterized by calcification and thickening of the aortic valve leaflets, resulting in a narrowed orifice and increased pressure gradient across the valve. AS typically progresses from a subclinical phase known as aortic sclerosis, where valve calcification occurs without a transvalvular gradient, to a more advanced stage marked by a triad of symptoms: heart failure, syncope, and angina. Echocardiography plays a crucial role in the diagnosis and evaluation of AS, serving as the primary non-invasive imaging modality. However, to minimize misdiagnoses, it is crucial to adhere to a standardized protocol for acquiring echocardiographic images. This is because, despite continuous advances in echocardiographic technology, diagnostic errors still occur during the evaluation of AS, particularly in classifying its severity and hemodynamic characteristics. This review focuses on providing guidance for the imager during the echocardiographic assessment of AS. Firstly, the review will report on how the echo machine should be set to improve image quality and reduce noise and artifacts. Thereafter, the review will report specific emphasis on accurate measurements of left ventricular outflow tract diameter, aortic valve morphology and movement, as well as aortic and left ventricular outflow tract velocities. By considering these key factors, clinicians can ensure consistency and accuracy in the evaluation of AS using echocardiography.

## 1. Introduction

Aortic stenosis (AS) is the most prevalent valvular heart disease worldwide and represents a significant cause of cardiovascular morbidity and mortality [1,2,3,4]. Calcific AS leads to increased leaflet thickness and stiffness and to a narrowed aortic valve orifice, resulting in a pressure gradient across the valve [5,6]. AS has a prolonged subclinical phase known as aortic sclerosis, characterized by valve calcification in the absence of a transvalvular gradient [7]. As AS progresses, aortic valve narrowing, along with left ventricular pressure overload and subsequent left ventricular hypertrophy, give rise to the classic triad of AS symptoms: heart failure, syncope, and angina [1,8,9].

Echocardiography is the primary tool for diagnosing and evaluating AS, serving as the key non-invasive imaging method in this setting [10,11]. 

In order to ensure consistency and accuracy across echocardiography laboratories, a standardized approach should be followed while recognizing the inherent limitations of the method itself. This review aims to discuss considerations that the imager should be aware of during the echocardiographic evaluation of AS. Specifically, it focuses on the appropriate methods for the assessment of left ventricular outflow tract (LVOT) diameter, aortic valve morphology and movement, and aortic and left ventricular outflow tract velocities at transthoracic echocardiography.

Echocardiographic machine setting. The most important way to execute a correct echocardiogram in the evaluation of aortic stenosis, which is particularly prone to errors, is to correctly set your echo machine [12]. Generally, echocardiographic machines offer a variety of imaging modes and features. Pre-processing and post-processing are two important stages in echocardiographic imaging involving different operations to enhance image quality. Sonographers and doctors should participate actively in the setting of their echo machines.

Pre-processing represents what we cannot change after image acquisition. It involves:(1)Time-gain compensation (TGC) settings optimize the overall image brightness and compensate for variations in tissue depth. This helps in achieving a balanced image throughout the depth. Usually, the TGC curve is set flat (equal amplification for all depths), and thereafter it is adjusted as needed. It is well set by a) observing the image quality at different depths and adjusting the TGC settings to compensate for attenuation, and b) increasing the TGC gain gradually for deeper structures to compensate for signal loss. It is important to avoid excessive TGC compensation, as it can introduce noise and degrade image quality.(2)Filtering: The machine applies various filters, such as frequency filters or noise filters, to remove unwanted noise and artifacts from the acquired ultrasound data. Filtering helps in improving image quality and reducing interference. (A) A high-pass filter helps to eliminate motion artifacts caused by patient movement or cardiac motion. The exact setting for the high-pass filter depends on the desired balance between noise reduction and preservation of low-frequency information. (B) A low-pass filter helps to reduce speckle noise, which is a granular noise pattern inherent in ultrasound images, by eliminating high-frequency noise from the image. The low-pass filter setting should be adjusted to minimize noise while preserving important high-frequency details.(3)Persistence: Persistence is a temporal filtering technique allowing the image to retain previously displayed information for a certain duration. In echocardiography, high persistence can introduce artifacts and degrade image quality, first of rapidly moving structures. In addition, it can also introduce false motion artifacts, which can misrepresent the actual movement of the cardiac wall, valves, and chordae. Therefore, it is generally recommended to set the persistence level to a low or moderate value. This allows for real-time imaging with minimal trailing or image artifacts.(4)Harmonic imaging: Harmonic imaging utilizes the harmonic frequencies produced by tissue and blood, which can improve image quality by reducing noise and improving contrast. It is particularly useful for visualizing deeper structures and improving image clarity.(5)Frequency selection: Echocardiographic machines have different frequency settings for the ultrasound waves. Higher frequencies provide better resolution for superficial structures but have limited depth penetration. Lower frequencies offer better penetration but may compromise image quality. The appropriate frequency selection depends on the patient’s body habitus and the depth of the structure being examined.(6)Beamforming: Beamforming involves combining ultrasound signals from multiple elements of the transducer array to form the ultrasound beam. Beamforming determines the spatial resolution and image quality by focusing the ultrasound beam at different depths and angles.(7)Pre-processing includes operations such as signal amplification, dynamic range compression, and time-gating, which enhance the quality and clarity of the ultrasound signals. These processes optimize the visualization of cardiac structures and improve the accuracy of measurements.(8)Motion compensation: This method helps in stabilizing the image and enhancing the visibility of structures by frame averaging or frame interpolation.(9)Transducer selection: Echocardiographic machines come with different transducers designed for specific purposes. The choice of transducer depends on the patient’s body habitus (such as body size), the area of interest, and the depth of penetration required.(10)Depth and width of the image: The depth setting determines the imaging depth within the patient’s body. It should be adjusted according to the structure of interest. Deeper structures may require a larger imaging depth to ensure proper visualization, while shallower structures can benefit from a smaller imaging depth. This is the same for the width setting. It should be remembered that the frame rate reduces at increasing depth and width.

Post-processing involves:(1)The reconstruction of 2D images from the acquired ultrasound data. The machine applies algorithms to create a visual representation of the heart and its structures.(2)Image enhancement: Post-processing offers image sharpening, contrast adjustment, and edge enhancement. These tools help to further improve image quality, highlight important structures, and aid in the diagnostic process.(3)Image manipulation: Image manipulation includes rotation, cropping, zooming, and multi-planar reconstruction. These capabilities enable the user to view the acquired images from different angles and perspectives for a more comprehensive evaluation.(4)Gain adjustment: The gain controls the amplification of the returning ultrasound signals. It is important to optimize the gain settings to achieve optimal image quality. Excessive gain can result in a bright image with noise, while insufficient gain can make the image too dark. Adjusting the gain appropriately ensures optimal visualization of cardiac structures (see later).

All the settings, pre- and post-processing, are reported in Figure 1.

Now, sonographers and/or doctors are ready for image acquisition.

## 2. Long Axis View

Transthoracic echocardiography should be initiated with the long-axis view [12]. From this view, we can measure the left ventricular outflow tract (LVOT) and begin assessing the morphology and movement of the aortic valve [13,14]. Importantly, sonographers should optimize the long axis as more perpendicular to the aortic posterior wall and to the anterior septum so that the lowest angle between the aortic wall and the anterior septum is reached. Figure 2 reports what should be avoided during long-axis acquisition.

LVOT diameter. The accurate two-dimensional (2D) measurement of LVOT is crucial when assessing AS, as it represents a significant potential source of error in calculating the aortic valve area (AVA) using the continuity equation [15,16,17]. The LVOT measurement is squared within the continuity equation, so any small error in measurement is significantly amplified [18,19]. LVOT diameter should be measured in mid-systole [20]. Mid-systole is the point of the cardiac cycle when the shape of the aortic annulus becomes most circular, and its area becomes maximal. However, for the LVOT, there is no evidence of shape variation from oval to circular throughout systole. Nonetheless, mid-systole should be identified as the frame at half of the total number of frames available between the aortic opening and closure. In this mid-systolic frame, LVOT is measured from the inner edge to the inner edge of the septal endocardium to the anterior mitral leaflet, parallel to the plane of the aortic valve [20]. If a long-axis view is not available, it is not recommended to measure the LVOT from apical views. It is important to recognize that during image acquisition, the parasternal long-axis view provides better axial resolution, whereas the LVOT apical view relies on lateral resolution, which is a limiting factor. It is important to recognize that after the identification of a correct long axis, imaging resolution improvement is fundamental. Axial resolution, as said before, is the most precise of the other two resolutions, lateral and elevational. Thus, quantitative measurements are made most reliably using data derived from a perpendicular alignment. In order to improve axial resolution, higher frequencies should be used. In addition, harmonic imaging is always recommended in that it improves endocardial definition [21]. Gain controls, which adjust the displayed amplitude of the received signals, should be corrected because excessive gain settings can cause “blooming” of the echoes (Figure 3, left panel), leading to a significant underestimation of the LVOT diameter. Conversely, by lowering too much, the gain prevents proper visualization of the anterior endocardium, resulting in an overestimation of the LVOT diameter (Figure 3, right panel). 

Calcium, often observed within the LVOT at the level of the mitral anterior leaflet, should be excluded (Figure 4). 

At this point, the zoom mode is recommended. Can we now measure LVOT? Not yet. Recently updated guidelines from the American Society of Echocardiography suggest measuring 3–10 mm below the annulus level [20]. The rationale behind measuring 3–10 mm below the annulus is based on the requirement for the pulsed Doppler sample volume to be in the same anatomic plane where the cross-sectional area is calculated for accurate continuity equation results. Therefore, when obtaining the LVOT velocity in the presence of AS, if the Doppler cursor is backed away from the aortic valve, to avoid pre-valvular acceleration, then the location of 2D LVOT measurement in parasternal long must also be backed away from the aortic valve as well [22]. 

Aortic morphology and movement. From the long axis, aortic valve morphology and movement could be assessed; however, in general, a short axis is additionally required. When stenosis is present, 2D images show a marked increase in echogenicity consistent with calcification of the leaflets. A systolic leaflet separation of at least 15 mm by 2D or 2D-guided M-Mode reliably excludes severe stenosis [23]. If less than 15 mm, the degree of stenosis can range from mild to severe (Figure 5.). 

Of note, systolic doming from the long-axis view is a clue of a bicuspid valve.

Short axis view. The short axis view visualizes the three aortic leaflets. However, planimetry of the valve may be misleading because the aortic orifice is not planar. From this view, the bicuspid, sclerotic, and calcified aortic valve is diagnosed. In addition, imaging setting optimization, reported for the long-axis view, must also be performed in this view (Figure 6).

A bicuspid valve most often results from fusion of the right and left coronary cusps, so a larger anterior and smaller posterior cusp with both coronary arteries arising from the anterior Valsava sinus is diagnosed in 80% of cases, whereas fusion of the right and non-coronary cusps resulting in larger right than left cusp, with one coronary artery arising from each Valsava sinus, is less common (20% of cases) [24,25,26,27] (Figure 7). 

The short-axis views of the valve in systole should image a typical ‘‘fish-mouth’’ appearance of valve opening and absence of opening at the raphe. 

Echocardiographically, aortic sclerosis is defined by focal areas of valve thickening, typically located in the leaflet center or base, with commissural sparing and normal leaflet mobility. With aortic sclerosis, valvular hemodynamics are within normal limits, with an antegrade velocity across the valve < 2.5 m/s [23,28,29]. The more extensive area of valve thickening is defined as calcification, which is usually present at the center or at the base of the leaflets, in the case of degenerative etiology [30]. In rheumatic aortic valve disease, similar to the mitral valve, and especially in more advanced stages, there is commissural fusion, with increased echogenicity along the leaflets edge. However, often rheumatic and degenerative calcific AS have similar imaging [31]. Calcification of a bicuspid valve is somewhat more asymmetric. In the setting of calcification with reduced leaflet excursion in systole, the abnormal motion of the cusps may not be appreciated, and color Doppler may be helpful in distinguishing immobile trileaflet aortic valves without commissural fusion from bicuspid valves with fusion; color Doppler flow in all three commissures should be seen with trileaflet valves [25,32,33]. The severity of valve calcification can be graded semi-quantitatively as mild (few areas of dense echogenicity with little acoustic shadowing), moderate (multiple larger areas of dense echogenicity), or severe (extensive thickening and increased echogenicity with a prominent acoustic shadow) [34,35,36]. Although echocardiography remains the first-line diagnostic technique, in up to 40% of patients, assessments between two or more observers lead to discordant results, leading to uncertainty on the true severity of the disease. In these cases, cardiac tomography for the quantification of aortic valve calcium has emerged as a valuable and complementary marker of aortic stenosis severity [37,38,39,40].

Pulsed Doppler. Pulsed Doppler represents blood flow velocities from a specific cardiac depth, with a depth of interest named sample volume. It is important to standardize the sample volume in your machine [21]. Usually, a sample volume length of 3–5 mm is recommended to obtain the best relationship between resolution and quality of the signal [21]. As pulsed Doppler uses the same piezoelectric crystals to send and receive sound waves, there is a maximum limit of the Doppler frequency that can be measured. If velocity is higher than the Nyquist limit, aliasing occurs, i.e., an apparent change in the direction of blood flow [21]. High pulse repetition frequencies are not suggested because it has the drawback of including signals from each of the originated sample volumes. The gain and scale settings should be optimized in such a way that the entire curve can be seen. The machine operator must ensure that the Doppler signal is large enough to measure accurately the blood velocity in that specific cardiac depth; the velocity curve should be smooth and have a dense outer edge to be measured, and the small, fine, linear signals should be excluded. LVOT spectral Doppler velocity time integral (VTI) assessment with the sample volume in a wrong position or being too large, with too high gain or too high wall filter settings, low sweep speed, and baseline inappropriately low [22,41]. The LVOT VTI is recorded from an apical approach, using either an anteriorly angulated four-chamber view or positioning the sample volume of pulsed Doppler proximal to the region of blood flow acceleration in the LVOT (Figure 8).

Under color Doppler guidance, the sample volume should be initially placed 1 cm proximal to the aortic valve while recording the velocity. Then, the sample volume should be moved toward the aortic valve until an increase in velocity and spectral broadening is seen; thereafter, the sample volume is slowly repositioned apically with a progressive reduction in velocity and spectral broadening to obtain the best possible envelope quality. The presence of an aortic valve closing (not opening) click, when visualizable, suggests a proper positioning of the sample volume in the LVOT. Wall filters should be set low enough that the systolic ejection period is clearly defined. The pulsed Doppler settings should be optimized for accurate LVOT VTI estimation: wall filters should be set to low levels and the gain reduced until the brightest (or densest) portion of the spectral tracing is seen, known as the ‘modal velocity’, which represents the velocity of most blood cells [22,42]. The outer edge of the modal velocity should be traced for the LVOT VTI. Sweep speed should be reduced to make area assessment more accurate. As for the LVOT diameter, two or three cardiac cycles should be averaged for a patient in sinus rhythm and from five to seven for a patient in atrial fibrillation. Once the distance from the aortic valve at which we recorded the VTI of the outflow tract is identified, we need to go back to the long axis and measure the diameter of LVOT at the same distance.

Continuous Doppler. Aortic maximal velocities are squared in the Bernoulli equation, meaning that if the Doppler is poorly aligned and underestimated, this error will also be squared in the calculation [14,43,44]. Moreover, if the Doppler is mismeasured by incorrect tracing, this error will result in another squared error. 

When measuring, it is important to search not only the highest velocities but also note which acoustic window yields this peak AS jet velocity [29,45]. Attention should be paid to identifying the highest velocity, i.e., aliasing, seen at color Doppler, not only at valve level but also in the ascending aorta so that the continuous beam should be placed as parallel as possible to this flow (Figure 9).

Considering that the direction of the aortic jet is often eccentric relative to both the plane of the aortic valve and the long axis of the aorta, in the effort to search for a near parallel angle between the ultrasound beam and the direction of the jet, one should use several acoustic windows with a proper positioning of the patient and multiple transducer angulations [46]. In the beginning, the aortic jet should be interrogated from the apical approach, with the patient in the left lateral decubitus, as the patient should be analyzed by the high right parasternal border, with the subject on a steep right lateral decubitus, and from the suprasternal notch, with the patient supine and the neck slightly extended over a pillow. Proper patient positioning is key, as well as ensuring Doppler alignment is parallel with AV antegrade flow. Respiratory variations while imaging should also be avoided; having the patient perform inspiratory or expiratory with breath-holding maneuvers helps to ensure proper CWD alignment [47,48]. If a small dual-crystal Doppler transducer (PEDOF-probe) is available, it is recommended to use it because of its smaller footprint and ability to detect signals better between intercostal spaces and in the little window of the suprasternal view. A helpful tip is to try utilizing the imaging probe first to ensure proper alignment and the highest velocities, maintain that position, and then swap out to the PEDOF-probe so the “blind hunt” for the velocity is less laborious [49]. This can be performed in all the recommended acoustic windows. Of note, the apical and right sternal border typically yield the highest velocities.

Figure 10 synthesizes the overall steps suggested in the present review.

Post transcatheter aortic valve implantation (TAVI)) or surgical aortic valve replacement (SAVR) measurements.

When assessing patients with aortic prosthesis, documentation of the size and type of valve should be marked on all echocardiograms. In addition, it is important to know, often for the aortic prosthesis, the level at which the prosthesis has been implanted: intra-annular, partially supra-annular, or wholly supra-annular. These aspects are important, since each valve type and size have their own normal ranges of values, such as peak velocity, peak gradient, mean gradient, and aortic valve effective orifice area (EOA). If repeat echocardiograms are being performed on a patient that has a prosthetic AV, the prior echocardiogram report should be reviewed to see what the AV measurements were, so any changes in function or abnormalities can be caught. To study aortic prosthesis, we will apply the same approach and calculations as we applied for a native valve. The prosthesis should be imaged from multiple views, with particular attention to (1) the opening and closing motion of the moving parts of the prosthesis (leaflets for bioprostheses and occludes for mechanical prostheses), (2) the presence of leaflet calcifications or abnormal echo density attached to the sewing ring, occluder, leaflets, stents, or cage, and (3) the appearance of the sewing ring, including careful inspection for regions of separation from the native annulus and for abnormal rocking motion during the cardiac cycle [50]. An important difference between the native aortic valve and prosthesis relies on the LVOT assessment. The LVOT diameter should now be measured at the ventricular side of the prosthesis, from outer edge to outer edge of the stent or ring just below the sewing ring for surgical prostheses or the stent for transcatheter bioprostheses [51] (Figure 11). 

Recall that the LVOT pulsed-wave sample volume should be placed at the same level in the apical view that provides the best alignment with outflow through the device [52]. 

The correct assessment of LV systolic, diastolic function, and hypertrophy, and of aortic root dimensions.

In addition to assessing the presence and severity of aortic stenosis (AS), echocardiography should measure other parameters to determine the feasibility of treatments and predict outcomes. Some of the most important parameters include LV ejection fraction, diastolic function, LV hypertrophy, and aortic root dimensions. LV ejection fraction, such as patients with myocardial infarction [53], is an important prognostic marker in AS [54]. Proper alignment of the apical view is crucial to avoid apex shortening and obtain a maximal longitudinal LV axis [55,56]. 

Assessing diastolic function using Doppler ultrasound may be challenging due to the presence of mitral annular calcification, which can impede tissue Doppler imaging [57,58]. 

Left atrial (LA) size can serve as a surrogate for diastolic function and increases with the burden of diastolic pressures [59]. LA size can be assessed using the long-axis view, linear dimensions, or apical 4- and 2-chamber views (volumetric dimension). Volumetric assessment is recommended [55], as small variations in linear dimensions can result in significant volume variations [60]. Algorithms have been proposed to estimate LA volume from linear dimensions in case LA volume is not available [61]. By applying this algorithm, at increasing LA size, the prognosis worsens in patients with mild to moderate AS and with LV ejection fraction ≥40% [62]. 

A higher LV mass index is independently associated with increased cardiovascular morbidity and mortality during the progression of aortic stenosis [63]. In order to assess LV mass, a correct long-axis view is recommended for measuring the LVOT as it allows for direct linear measurements; electronic calipers should be placed at the interface between the myocardial wall and cavity, as well as the interface between the wall and pericardium [55,64]. 

A small aortic root frequently is present together with a small aortic annulus; both have been associated with poorer outcomes after aortic valve replacement, with increased risk of mortality, ischemic cardiovascular events, and stroke [65]. The aortic root dimension should be obtained from the parasternal long-axis view, with the transducer closer to the sternum and slightly oriented to the right shoulder of the patient. The obtained long axis depicts the aortic root and the proximal ascending aorta [66]. The leading-edge to leading-edge method is the most accepted method to aortic root measurement [67,68,69].

## 3. Conclusions

Echocardiography is the gold standard for assessing valve disease. In the case of aortic stenosis, the role of 2D and Doppler echocardiography is well established. However, the degree of stenosis can vary depending on the imaging acquisition in the same patient. Reducing this variability is important for standardizing image acquisition and processing, which will also aid in the follow-up of patients. This is particularly crucial for patients with calcific aortic stenosis, as the severity of stenosis can progress rapidly.

## Figures and Tables

**Figure 1 diagnostics-13-02414-f001:**
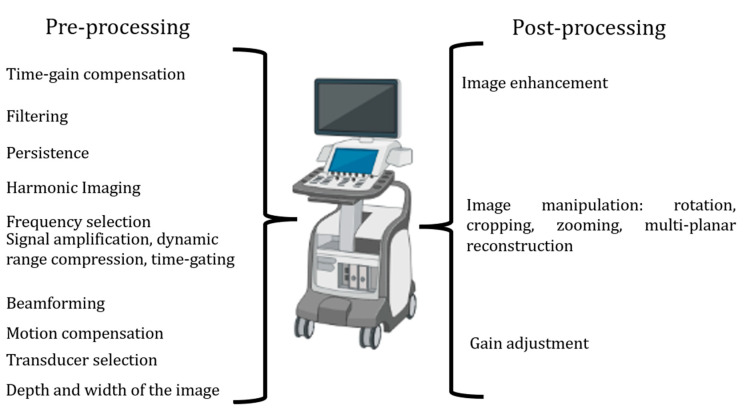
The steps of pre-processing and post-processing to be set to improve image quality.

**Figure 2 diagnostics-13-02414-f002:**
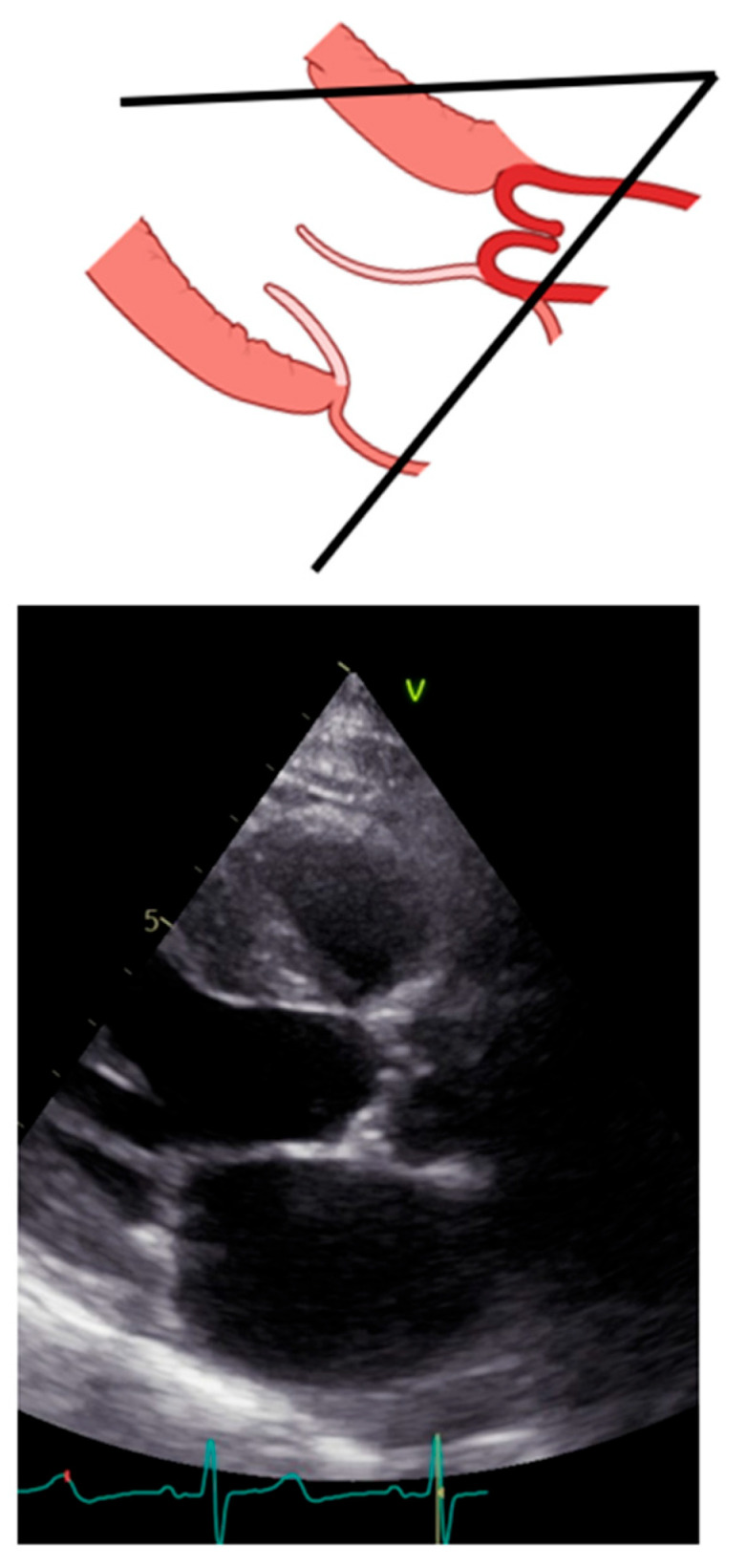
Incorrect long-axis view.

**Figure 3 diagnostics-13-02414-f003:**
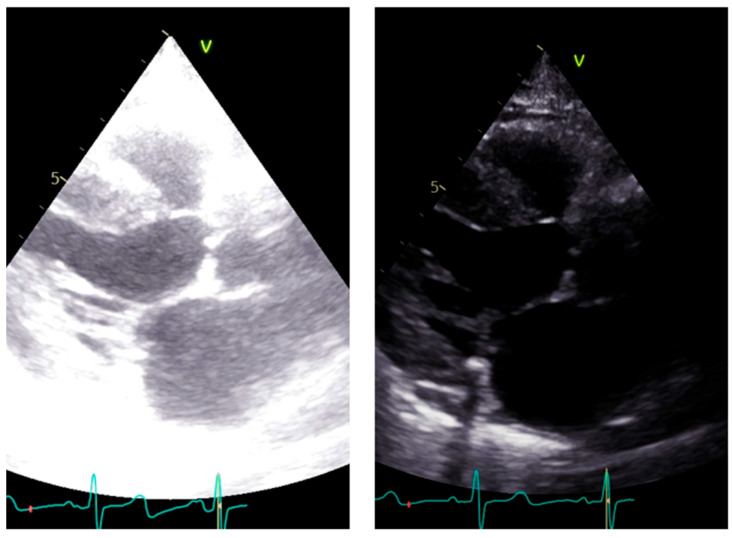
Over gain (**left panel**) and lower gain (**right panel**) in the long-axis view along the same patient.

**Figure 4 diagnostics-13-02414-f004:**
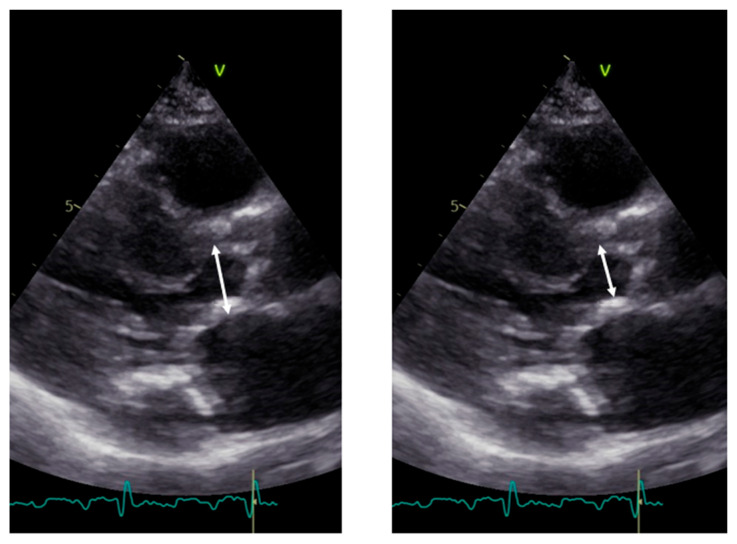
Measurement of LVOT in the presence of calcium. **Left panel**: incorrect measurement including calcium; **right panel**: correct measurement excluding calcium.

**Figure 5 diagnostics-13-02414-f005:**
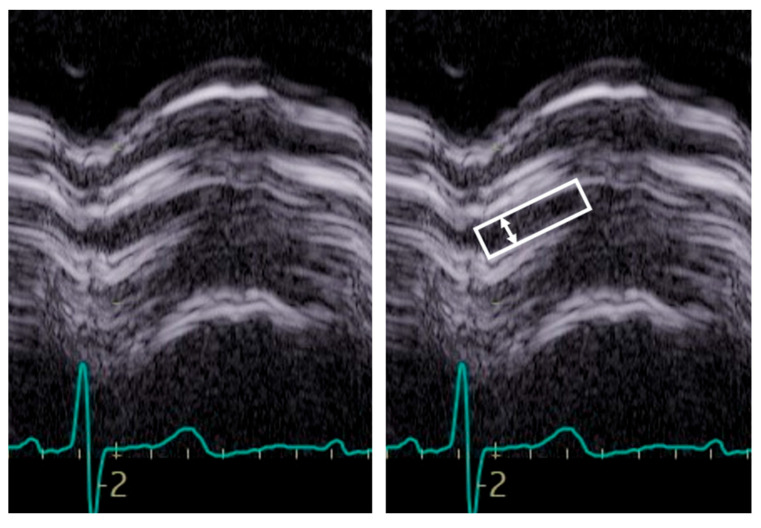
Example of M-Mode tracing of the aortic valve. Note the reduced opening, inferior to 15 mm, suggestive of aortic stenosis although not of a specific degree. The box indicates the aortic valve opening, the arrow the maximal opening.

**Figure 6 diagnostics-13-02414-f006:**
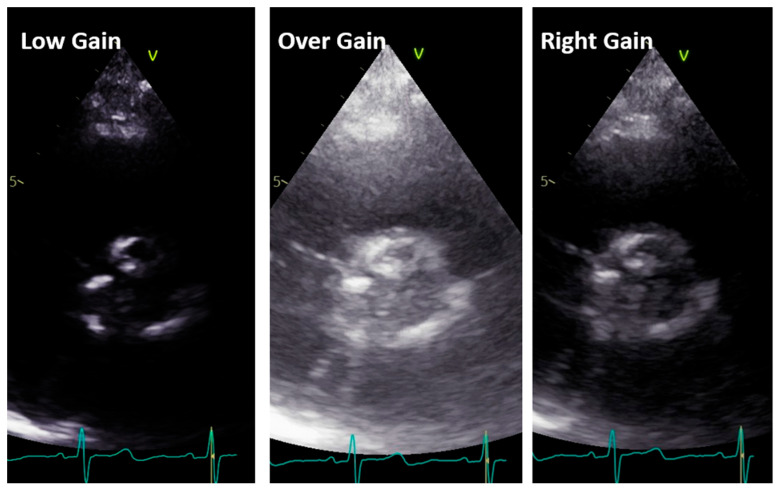
Low gain (**left panel**), over gain (**middle panel**), and right gain (**right panel**) in the short-axis view along the same patient.

**Figure 7 diagnostics-13-02414-f007:**
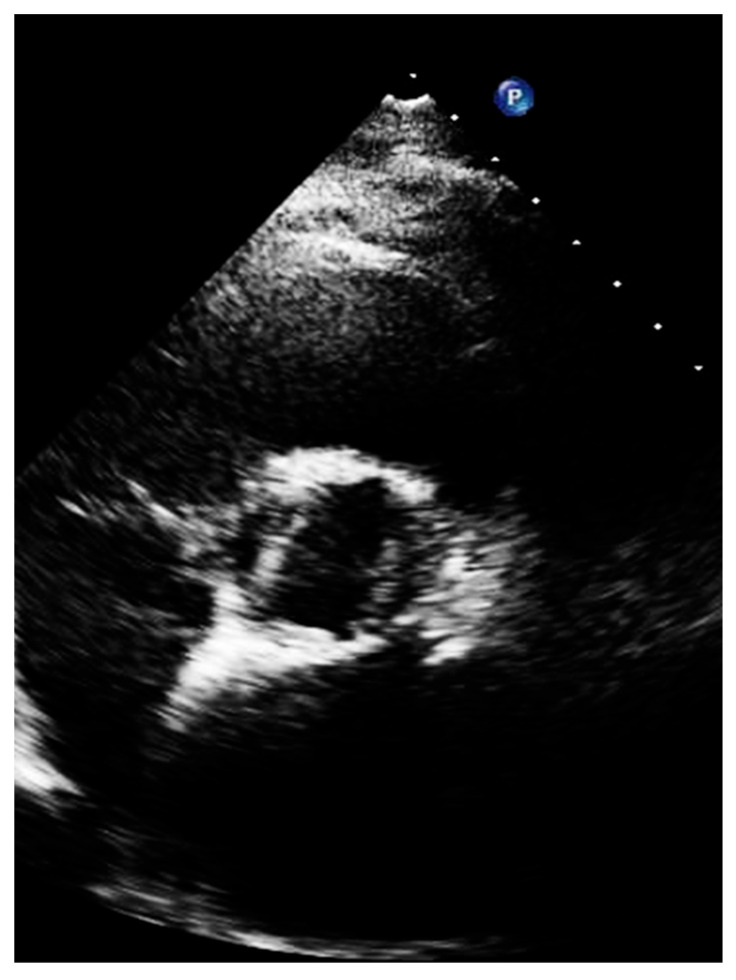
Short axis of a patient with bicuspid aortic valve, with large left and right cusp.

**Figure 8 diagnostics-13-02414-f008:**
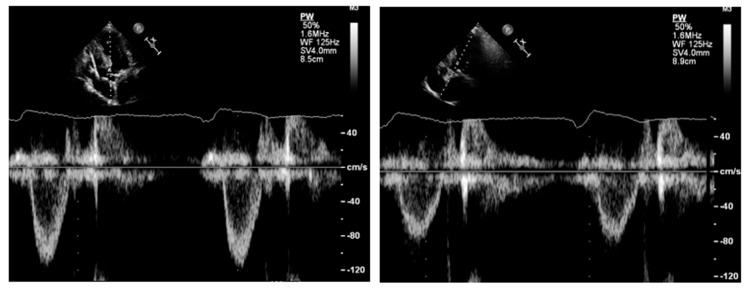
**Left panel**: underestimation of left ventricular outflow velocity time integral due to increased angulation between the sample volume and the direction of blood flow. **Right panel**: reduction in the angle results in an increase in the left ventricular outflow tract velocity time integral.

**Figure 9 diagnostics-13-02414-f009:**
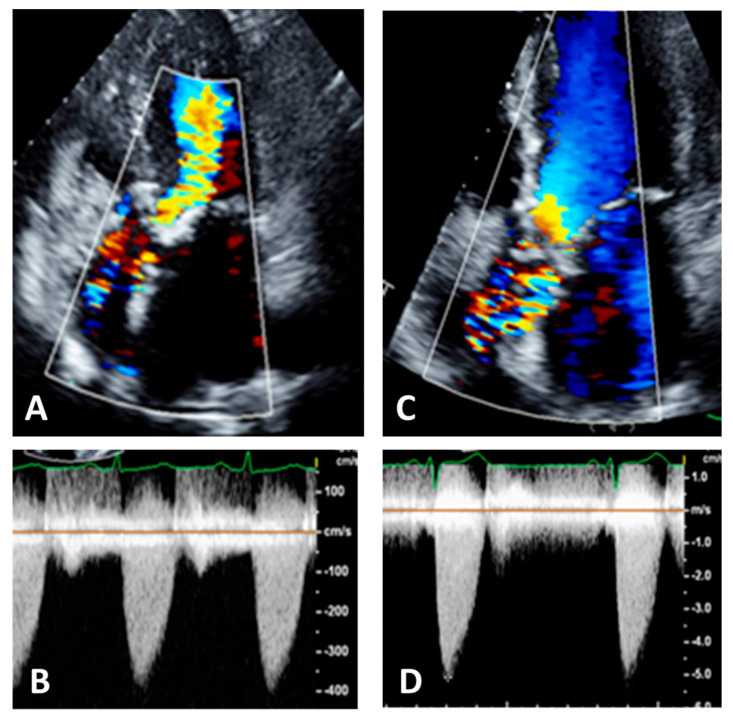
Panel (**A**): five chamber view. Color Doppler is not well visualized in ascending aorta, Panel (**B**); this results in an underestimation of the aortic velocities. Panel (**C**): the same patient with correct visualization of color Doppler in ascending aorta, Panel (**D**). This results in an increase and correct recording of aortic velocities.

**Figure 10 diagnostics-13-02414-f010:**
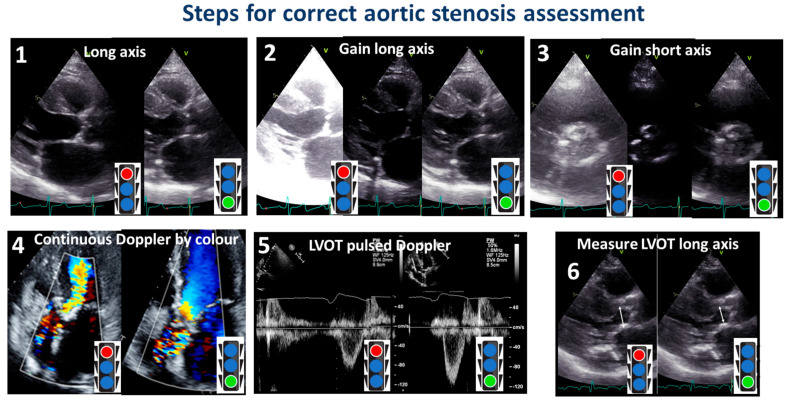
Steps for a correct assessment of aortic stenosis. The red color of the stoplight represents what should be avoided, whereas the green color represents what should be done. LVOT: left ventricular outflow tract.

**Figure 11 diagnostics-13-02414-f011:**
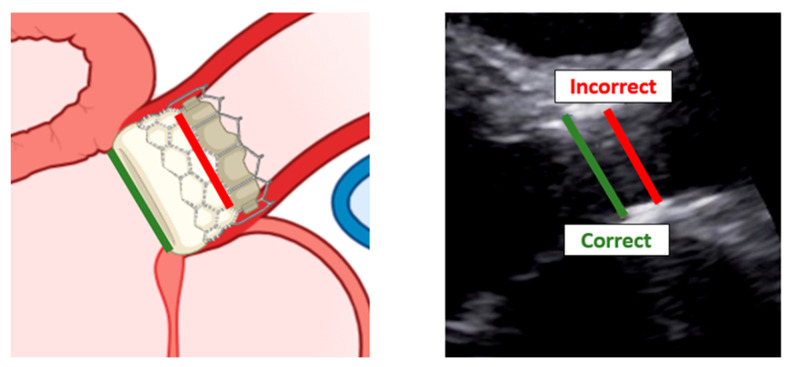
Correct measurement of left ventricular outflow tract in a patient with trans-catheter aortic valve implantation. **Left**: scheme of a left ventricular outflow tract. **Right**: a long-axis view with the implanted bioprostheses.

## Data Availability

Not applicable.

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
