# Peer review of "Pitfalls and Tips in the Assessment of Aortic Stenosis by Transthoracic Echocardiography"

_diagnostics, 2023, doi:10.3390/diagnostics13142414_

Round 1

Reviewer 1 Report

In this manuscript the authors review and comment on echocardiographic assessment of aortic stenosis. Although the topic is not novel, but rather well-documented and studied, the manuscript provides valuable remarks for echocardiographers. Minor suggestions:

lines 65-72: although time gain compensation cannot be modified after examination, overall gain can, thus it may be more appropriate to include gain only  in post-processing, as already done (lines 123-127), and not in pre-processing.

lines 128-133: depth and sector width are defined before image acquisition and cannot be changed after acquisition (apart from cropping, but this does not actually change the initial settings), thus they may be included in the pre-processing parameters. They are of critical importance, as they determine both frame rate and scan line density and this should be mentioned.

line 150: figure legend corresponds to the first part of the figure, not the echo image. As far as I can see, the 2 echo images are from the same patient, from slightly different phases of the cardiac cycle and this creates some confusion. Thus, they do not clearly indicate the difference that the authors want to underline, and in my opinion they should be substituted by a more representative example or completely omitted (keep only the first part of the figure)

line 166: It is important to realize that, during image acquisition,...

lines 229-230: please correct the typo in the legend (over gain: middle, Low gain: left)

lines 232-245: regarding coronary artery ostia, a reference to Valsalva sinuses rather than aortic cusps should be made.

line 257: in the case of degenerative etiology. In rheumatic aortic valve disease,...

line 259: especially in more advanced stages

line 269: what do the authors mean by saying discordant? Do they mean low-flow low gradient severe AS? Or that AS may be severe in one echo exam and moderate in another? Or maybe discordance between 2D data and doppler hemodynamics? This should be further clarified.

line 278: doppler shift or frequency shift or blood velocity instead of frequency

line 279: Nyquist limit or this limit instead of repetition frequency

lines 280-281: High pulse repetition frequency has the drawback of...

lines 285-287: what do the authors mean?

lines 331-347: in my experience, suprasternal view for aortic velocity determination is only useful with a PEDOF probe.

line 22: advances instead of advancements

line 134: and instead of e

line 155: ...so any small error in measurement is signifiantly amplified

lines 156-158: ...Mid-systole is the point of cardiac cycle when the shape of the aortic annulus becomes most circular and its area becomes maximal.

line 175: too much

line 205: suggest

line 214: in general,...

line 222: Of note instead of In addition

line 260: somewhat instead of sometime

line 275: recommended instead of suggested

line 277: receive instead of analyze

line 293: increased

line 294: omit correctly

line 339: as

Author Response

We appreciate the thoughtful comments and valuable suggestions to improve impact of our paper. In this letter we used black for reviewer’s comments and black for our answers

In this manuscript the authors review and comment on echocardiographic assessment of aortic stenosis. Although the topic is not novel, but rather well-documented and studied, the manuscript provides valuable remarks for echocardiographers.

We would like to thank the reviewer for her/his consideration in our manuscript.

Minor suggestions:

lines 65-72: although time gain compensation cannot be modified after examination, overall gain can, thus it may be more appropriate to include gain only  in post-processing, as already done (lines 123-127), and not in pre-processing.

We deleted gain in the preprocessing.  Accordingly, we changed Figure 1.

lines 128-133: depth and sector width are defined before image acquisition and cannot be changed after acquisition (apart from cropping, but this does not actually change the initial settings), thus they may be included in the pre-processing parameters. They are of critical importance, as they determine both frame rate and scan line density and this should be mentioned.

We would like to thank the reviewer for her/his attention. We reported deph and sector in the pre-processing sub-heading. Accordingly, we changed Figure 1

line 150: figure legend corresponds to the first part of the figure, not the echo image. As far as I can see, the 2 echo images are from the same patient, from slightly different phases of the cardiac cycle and this creates some confusion. Thus, they do not clearly indicate the difference that the authors want to underline, and in my opinion they should be substituted by a more representative example or completely omitted (keep only the first part of the figure).

As suggested, wee kept only the first part of the figure

line 166: It is important to realize that, during image acquisition,...

We added the sentence.

lines 229-230: please correct the typo in the legend (over gain: middle, Low gain: left).

Thank you. We corrected the typos.

lines 232-245: regarding coronary artery ostia, a reference to Valsalva sinuses rather than aortic cusps should be made.

We corrected accordingly to your suggestion.

line 257: in the case of degenerative etiology. In rheumatic aortic valve disease,...

We corrected accordingly to your suggestion.

line 259: especially in more advanced stages

We corrected accordingly to your suggestion.

line 269: what do the authors mean by saying discordant? Do they mean low-flow low gradient severe AS? Or that AS may be severe in one echo exam and moderate in another? Or maybe discordance between 2D data and doppler hemodynamics? This should be further clarified.

The reviewer is right. The discordance refers to different diagnosis among the same patient by different echocardiographers. We rephrased accordingly: Although echocardiography remains the first line diagnostic technique, in up to 40% of patients, assessments between two or more observers leave to discordant results, leading to uncertainty on the true severity of disease.

line 278: doppler shift or frequency shift or blood velocity instead of frequency

We corrected accordingly to your suggestion.

line 279: Nyquist limit or this limit instead of repetition frequency

We corrected accordingly to your suggestion.

lines 280-281: High pulse repetition frequency has the drawback of...

We corrected accordingly to your suggestion.

lines 285-287: what do the authors mean?

Sorry the sentence was not complete, now is understandable: The machine operator must ensure that the Doppler signal is large enough to measure ac-curately the blood velocity in that specific cardiac deph.

lines 331-347: in my experience, suprasternal view for aortic velocity determination is only useful with a PEDOF probe.

Even in our experience, so we corrected a sentence that include the use utility of Pedof by suprasternal window. If small dual-crystal Doppler transducer (PEDOF-probe) is available, it is recommended to use it because of its smaller footprint and ability to detect signals better between intercostal spaces and in the little window of the suprasternal view.

Comments on the Quality of English Language

line 22: advances instead of advancements

line 134: and instead of e

line 155: ...so any small error in measurement is signifiantly amplified

lines 156-158: ...Mid-systole is the point of cardiac cycle when the shape of the aortic annulus becomes most circular and its area becomes maximal.

line 175: too much

line 205: suggest

line 214: in general,...

line 222: Of note instead of In addition

line 260: somewhat instead of sometime

line 275: recommended instead of suggested

line 277: receive instead of analyze

line 293: increased

line 294: omit correctly

line 339: as

We corrected accordingly to overall suggestion.

Reviewer 2 Report

I would like to congratulate authors for this article . But it will need some more details of assessment.

1) There is no mention of Left ventricular hypertrophy assessment. As most of the time Severe Aortic stenosis has associated LVH, this has to be mentioned.

2) Left Ventricular Systolic & Diastolic function is integral part of Aortic stenosis assessment is not mentioned. The entities like Low Flow Aortic stenosis etc needs to be described in detail.

3) Assessment of Aortic root and arch is not mentioned. This is important before planning surgery in Severe Aortic stenosis.

4) Description of Figure 6 needs to be corrected.

I am sure addition of these will make this article complete.

Author Response

 I would like to congratulate authors for this article .

We would like to thank the reviewer for her/his consideration in our manuscript.

But it will need some more details of assessment.

1) There is no mention of Left ventricular hypertrophy assessment. As most of the time Severe Aortic stenosis has associated LVH, this has to be mentioned.

2) Left Ventricular Systolic & Diastolic function is integral part of Aortic stenosis assessment is not mentioned. The entities like Low Flow Aortic stenosis etc needs to be described in detail.

3) Assessment of Aortic root and arch is not mentioned. This is important before planning surgery in Severe Aortic stenosis.

We added the following heading to answer to your points. The correct assessment of LV systolic, diastolic function and hypertrophy, and of aortic root dimensions.

In addition to assessing the presence and severity of aortic stenosis (AS), echocardiography should measure other parameters to determine the feasibility of treatments and predict outcomes. Some of the most important parameters include LV ejection fraction, diastolic function, LV hypertrophy, and aortic root dimensions. LV ejection fraction, like patients with myocardial infarction [53], is an important prognostic marker in AS [54]. Proper alignment of the apical view is crucial to avoid apex shortening and obtain a maximal longitudinal LV axis [55,56].

Assessing diastolic function using Doppler ultrasound may be challenging due to the presence of mitral annular calcification, which can impede tissue Doppler imaging [57,58].

Left atrial (LA) size can serve as a surrogate for diastolic function and increases with the burden of diastolic pressures [59].LA size can be assessed using the long-axis view, linear dimensions, or apical 4- and 2-chamber views (volumetric dimension). Volumetric assessment is recommended [55], as small variations in linear dimensions can result in significant volume variations [60]. Algorithms have been proposed to estimate LA volume from linear dimensions, in case LA volume is not available [61]. By applying this algorithm, at increasing LA size, prognosis worsens in patients with mild to moderate AS and with LV ejection fraction ≥40% [62].

Higher LV mass index is independently associated with increased cardiovascular morbidity and mortality during progression of aortic stenosis [63]. To assess LV mass, a correct long-axis view is recommended for measuring the LVOT as it allows for direct linear measurements electronic calipers should be placed at the interface between the myocardial wall and cavity, as well as the interface between the wall and pericardium [55,64].

A small aortic root frequently is present together with small aortic annulus; both have been associated with poorer outcomes after aortic valve replacement, with increased mortality, ischemic cardiovascular events, and stroke [65]. Aortic root dimension should be obtained from the parasternal long-axis view, with the transducer closer to the sternum and slightly oriented to the right shoulder of the patient. The obtained long-axis depicts the aortic root and the proximal ascending aorta [66]. The leading leading-edge to leading-edge method is the most accepted method to aortic root measurement [67-69].

4) Description of Figure 6 needs to be corrected.

We corrected the legend of the figure 6.

I am sure addition of these will make this article complete.

We hope now the paper is complete.